# Calcium Phosphate-Based Biomaterials for Bone Repair

**DOI:** 10.3390/jfb13040187

**Published:** 2022-10-14

**Authors:** Xiaodong Hou, Lei Zhang, Zifei Zhou, Xiong Luo, Tianlong Wang, Xinyu Zhao, Bingqiang Lu, Feng Chen, Longpo Zheng

**Affiliations:** 1Center for Orthopaedic Science and Translational Medicine, Department of Orthopedics, Shanghai Tenth People’s Hospital, Tongji University School of Medicine, Shanghai 200072, China; 2Department of Orthopedics, First Affiliated Hospital of Kunming Medical University, Kunming Medical Uni-versity, Kunming 650032, China; 3Shanghai Trauma Emergency Center, Shanghai 200072, China; 4Orthopedic Intelligent Minimally Invasive Diagnosis & Treatment Center, Shanghai Tenth People’s Hospital, Tongji University School of Medicine, Shanghai 200072, China

**Keywords:** calcium phosphate, bone regeneration, hydroxyapatite, biomineralization, osteogenesis

## Abstract

Traumatic, tumoral, and infectious bone defects are common in clinics, and create a big burden on patient’s families and society. Calcium phosphate (CaP)-based biomaterials have superior properties and have been widely used for bone defect repair, due to their similarities to the inorganic components of human bones. The biological performance of CaPs, as a determining factor for their applications, are dependent on their physicochemical properties. Hydroxyapatite (HAP) as the most thermally stable crystalline phase of CaP is mostly used in the form of ceramics or composites scaffolds with polymers. Nanostructured CaPs with large surface areas are suitable for drug/gene delivery systems. Additionally, CaP scaffolds with hierarchical nano-/microstructures have demonstrated excellent ability in promoting bone regeneration. This review focuses on the relationships and interactions between the physicochemical/biological properties of CaP biomaterials and their species, sizes, and morphologies in bone regeneration, including synthesis strategies, structure control, biological behavior, and the mechanisms of CaP in promoting osteogenesis. This review will be helpful for scientists and engineers to further understand CaP-based biomaterials (CaPs), and be useful in developing new high-performance biomaterials for bone repair.

## 1. Introduction

Bone as a mineralized tissue has an irreplaceable role in supporting and protecting the body of human beings. Defects of bone caused by trauma, aging, inflammation, infection, and tumors seriously affect people’s health and normal life [1]. A critical bone defect, which refers to a defect greater than 2 cm in length or greater than 50 percent of the circumference of the defect, cannot completely regenerate by self-growth and requires the use of biomaterials to guide its repair [2]. Millions of bone grafting operations are performed every year in the world for the treatment of critical bone defects, resulting in a huge economic burden to the families of patients and the whole of society [3]. Therefore, the development of high-performance biomaterials for bone repair is of great scientific significance and clinical application value.

At present, the most commonly used methods for treating bone defects include bone transplantation, membrane-guided regeneration, Ilizarov technology, and bone tissue engineering [4,5,6,7,8]. However, these methods are insufficient in meeting the requirements for clinical treatment of bone defects. Autologous bone grafting, the clinical gold standard for treating bone defects, is usually limited by the quantity of available tissue and the risk of secondary surgery and infection, blood loss and operation time [3]. Allografts and xenografts are important alternative options of autografts for orthopaedic applications in terms of osteogenic, osteoinductive, and osteoconductive properties, and there are two main categories, including cellular bone matrices (CBM) and peptide enhanced xeno-hybrid bone grafts developed as commercial products for clinical use. Cellular bone matrices have four necessary and beneficial elements for bone growth and healing: osteoinduction, osteoconduction, osteogenic activity and angiogenic activity [9]. However, CBM have several challenges with respect to intrinsic biological characteristics, such as viable cell sources, donor age at the time of graft harvest, and cell survival after transplantation, which may cause differences in expected outcomes due to different batches of the same product. Xeno-hybrid bone matrices are appealing, innovative, osteoconductive and osteoinductive bone substitutes to autografts. The compatibility of xeno-hybrid bone matrices in favoring cellular attachment, osseointegration, bone remodeling and satisfactory mechanical performance has been attested by numerous clinical studies [9]. However, further independent clinical studies are required to confirm these promising results and to promote their application. It is worth mentioning that although there are potential risks of infection when using allografts, the allografts are procured, processed, and distributed only by Tissue Banks, which operate under strict guidelines and sterile conditions in Class A environments, which helps to minimize the abovementioned issues [10]. The technology of tissue engineering represents an emerging strategy for repair of bone defect [11]. However, it is still a big challenge to construct functional bone tissue in vitro, due to the proliferation and differentiation of seeding cells, bioactivity of growth factors and physicochemical and biological properties of scaffolds [12,13]. Recently, in-situ tissue engineering has been proposed for autologous tissue regeneration, which is based on tissue-specific scaffolds, by regulating the microenvironment and in vivo recruiting stem and progenitor cells [14,15]. Therefore, the preparation of functional scaffolds with ideal biocompatibility, bioactivity and biodegradability is the critical factor that limits the rapid development of in-situ tissue engineering for bone defect repair.

Synthetic bone scaffolds have been increasingly applied in the field of bone repair. Compared with the autologous bone grafts, although there are some poor properties of osteinductive and osteogenic activities, synthetic bone scaffolds with abundant sources provide a wide variety of choices in structure, chemical/mechanical properties and biological functions to meet specific requirements in bone regeneration [15]. Considering the limitations above-mentioned, artificial bone substitutes have attracted tremendous attention and have been rapidly developed. Among the varied biomaterials used in bone repair, calcium phosphate (CaP)-based biomaterials occupy a particular position as a result of their resemblance to the chemical components and structures of natural bone tissue. CaP is not a specific material but represents a big family of materials that are compounds formed by the reaction of calcium ions and phosphate ions. The apatite reported by Werner in 1788 was the earliest discovered member of CaP [16]. By 1926, Jong revealed the relationship between apatite and the inorganic minerals of bone [17]. Therefore, CaP-based biomaterials (CaPs) were proposed for use as therapeutic agents for bone regeneration [18]. In 1971, Monroe was the first to report the use of CaP ceramics, which are white translucent polycrystalline ceramics that contain hydroxyapatite (HAP) [19]. Since that time, CaP ceramics have been developed greatly for the application of bone repair [20,21]. A CaP bone cement (CPC) was created by the hydrolysis of TCP and was used for the first time in the early 1920s as a treatment for bone repair [18]. Since then, CPC has been prepared with many different chemical formulas and application as described by Cama [22]. Mineralized collagen with orderly, organized collagen and HAP is the basic unit of natural bone tissues and is involved in building complex biomineralized systems with hierarchical structures [23]. Hence, researchers in the field of biomaterials are interested in biomimicking mineralized collagen for developing bone substitute materials by utilizing the biomimetics strategy [24,25]. In 2003, biomimetic mineralized collagen nanofibrils were designed and prepared by Cui et al., which are similar in both composition and structure to natural bone [26]. As of now, various methods have been developed for preparing mineralized collagen, and these products have excellent bioabsorbability and osteoconductive properties, leading to their potential in promoting bone regeneration [27].

With the advancement of materials science and technology, synthetic CaP materials, as promising biomaterials that provide similar bone environments for cell attachment, proliferation, and differentiation for bone regeneration, have attracted more and more interest and have shown excellent biocompatibility, osteoconductivity and osteoinductivity. However, synthesis methods, structural regulation and functionalization of CaPs are still long-term processes that need to be investigated to meet the requirements of different applications [28]. This review presents an up-to-date overview of advances in CaP biomaterials with different crystal phases and structures, strategies for fabricating biomimetic hierarchical nano-/microstructures and highlights their applications in bone regeneration (Figure 1 and Table 1). The review will help to further understand the relationships among the physical, chemical and biological properties of CaP biomaterials, and thus guide the preparation of the next generation of CaP biomaterials for bone repair.

## 2. Methods

The reports and information were gathered using the bibliographic databases PubMed and Web of Science. In addition, a search on Bing was performed to supplement the information. The following term combinations were searched: “calcium phosphate-based materials name” and (“bone regeneration” or “bone repair” or “bone healing”), “calcium phosphate-based materials name” including “calcium phosphate”, “CaP”, “hydroxyapatite”, “HA”, HAP”, “tricalcium phosphate”, “TCP”, “amorphous calcium phosphate”, “ACP”, “octacalcium phosphate”, “OCP”, “dicalcium phosphate anhydrous”, “DCPA”, “dibasic calcium phosphate dihydrate”, “DCPD”, “tetracalcium phosphate”, “TTCP”, “dicalcium phosphate monohydrate”, “DCPM”, “cement” and “ceramic”. Considering only innovative reports, the range of the publishing data was set as “2010–2022”. However, some classical and revolutionary works outside this window were also included. This study reviews and discusses representative literature.

## 3. Chemical Properties of Calcium Phosphate

### 3.1. Species of Calcium Phosphate

CaP is not only a family of natural minerals, but also includes biominerals in humans, which are the main inorganic component of hard tissue (bone and teeth) [31]. In the past decades, a wide variety of CaP-based biomaterials have been used in bone regeneration studies and clinical applications. As is well known, CaP biomaterials promote cell adhesion and growth, which then induces the formation of new bone minerals via their interaction with extracellular matrix proteins [43]. In the application of bone regeneration, the bioactivity of CaPs is critical and usually varied depending on their species [44]. The bioactive features of CaPs is related with to the degradation properties of CaP [45]. Due to different Ca/P ratios, the different species of CaP biomaterials result in variations in in vitro and in vivo calcium and phosphate ion release. Consequently, the pH of the local microenvironment of bone is affected by the released calcium and phosphate ions, which then influence the viability of osteoblasts and osteoclasts [46,47]. Moreover, the increased concentration of calcium and phosphate ions can promote the formation of bone minerals, as well as affect the expression of osteogenic differentiation-related genes (e.g., Col-I, ALP, OPN, OCN, RunX2 and BMPs) of bone cells [31,48].

Calcium exists wildly in natural bone minerals and is a key ion in forming the bone matrix [49]. Ca^2+^ is also capable of forming and maturing bone tissue by calcification. Ca^2+^ influence the bone cell maturation and bone tissue regeneration by regulating related cellular signaling pathways [50,51]. For instance, Ca^2+^ activating ERK1/2 causes pathway activation of osteoblastic-related bone formation [52]. In addition, an increased life span of osteoblasts has been observed with the activation of the PI3K/Akt signal axis by Ca^2+^ [53]. Meanwhile, phosphate ions degraded from CaP are present in large quantities in the human body, and can be utilized in various physiological systems, including construction of proteins, nucleic acids, and adenosine triphosphate [54]. Approximately 80% of phosphate ions in the body occur with calcium ions in the form of CaP in bone, which affects the development of bone tissue [55]. It is well known that the differentiation and growth of osteoblasts are regulated by phosphate ions by IGF-1, ERK1/2, BMP, and other pathways [56,57].

The osteoconductivity and osteoinductivity of CaP materials are closely related to their physical and chemical characteristics, such as, solubility, stability, and mechanical strength [31], are determined by the species of the CaP materials, Therefore, the selection of one kind of CaP biomaterial from their family according its characteristics is important in preparing biomaterials for the use of bone regeneration. A large number of CaP biomaterials have been used in bone regeneration and other biological research, including tricalcium phosphate (TCP), hydroxyapatite (HAP), amorphous calcium phosphate (ACP), octacalcium phosphate (OCP), dicalcium phosphate anhydrous (DCPA), dibasic calcium phosphate dihydrate (DCPD), and tetracalcium phosphate (TTCP) [42]. Basic information concerning these CaP biomaterials are displayed in the Table 2 [33]. The crystal phases of calcium phosphate were discovered before the 20th century (amorphous phases were discovered in the 1950s) and accurately characterized during the 20th century (OCP was defined in 1957 [58]). After that, no new crystal phases (non-doped, non-substituted, only Ca, P, O, H) have been reported. Dicalcium phosphate monohydrate (CaHPO_4_·H_2_O, DCPM) was obtained by controlling the transformation of a special amorphous calcium phosphate (CaHPO_4_·xH_2_O, ACHP) in a water-deficient environment (water/methanol mixed solvent, or in humid air) by Lu et al. in 2020 [59]. The discovery of DCPM brought a new member to the calcium phosphate family. However, applications of DCPM in bone repair, and even in biomedicine, has not yet been carried out.

### 3.2. Hydroxyapatite

HAP is the most abundant crystal phase of biominerals in human bones, and accounts for ~70% of the dry weight of bone tissue [60]. Among all CaP materials, HAP is only inferior to fluorapatite (FAP) in terms of stability and insolubility. The chemical formula of HAP is Ca_5_(PO_4_)_3_(OH). However, HAP is usually referred to as Ca_10_(PO_4_)_6_(OH)_2_ to indicate the hexagonal unit cell of HAP [61]. There are two approaches for the formation of HAP, including the natural formation process and artificial synthesis. The hexagonal crystal structure of naturally formed HAP usually has defects, which can be filled by vacancies or other ions [33]. However, structural defects in synthesized HAP may depend on synthesis procedures or conditions. Monoclinic and hexagonal crystals are the two crystal phases of synthesized HAP; the monoclinic crystal phase can change to the hexagonal crystal phase when the temperature is higher than 250 °C. The hexagonal crystal structure of HAP is the predominant phase found in the biological environment as a result of its high stability [62]. HAP is considered the most stable phase of CaP and the final mineral phase in bone, whereas the other CaP phases (e.g., ACP and OCP) in bone are precursors, or sub-precursors, that transform into HAP under in vivo or aqueous environments with high pH [31,33]. The phase transformation of several CaPs usually occurs under different conditions [63]. The equilibrium of phase transformation between various CaP phases is related to temperature and the ratio of CaO and P_2_O_5_. The physicochemical and biological properties of HAP significantly change with the Ca/P ratios and the replacement of ions or vacancies in HAP crystal structure. For example, the mechanical properties of HAP are enhanced with increasing Ca/P ratio, and reach a maximum when the stoichiometric ratio is 1.67. Interestingly, once the Ca/P ratio exceeds 1.67, the strength of HAP decrease [64]. The defects of HAP crystal structure can be replaced with F^−^, Cl^−^, CO_3_^2−^, Mg^2^^+^, Sr^2+^, and other ions. As a result of Mg^2+^ replacement, the size and density of HAP nanostructure particles, which contribute to the specific mechanical properties of bones, may be altered [65]. Furthermore, the crystallization of HAP is inhibited by Mg^2+^, and results in the formation of fewer large crystals and a greater number of apatite nuclei. The significance of this is that nanocrystalline bone apatites are necessary for the proper bone formation–resorption turnover that occurs via bone cells [66]. The replacement of F^−^ ions can increase stability, while Mg^2+^ enhances biological activity, compared with pure HAP [66]. Several studies have shown that Mg^2+^ has the ability to influence bone metabolism, regulate the activity of osteoblasts and osteoclasts, and to stimulate new bone growth [67,68]. Therefore, artificial Mg substituted HAP in different forms has been carried out, and has displayed advanced bioactivity [69,70]. Furthermore, Mg-based CaP materials can result in neuralization and the synthesis and release of CGRP to promote osteogenesis [71,72,73].

HAP has been used clinically in bone regeneration since the 1980s, as implants and coatings of other implants [61,74]. HAP has good biocompatibility, bioactivity, and osteoconductive properties. In the presence of Ca^2+^ and PO_4_^3−^ ions, the surface of HAP can act as a nucleation site for the initiation of biomineralization [75]. Therefore, HAP is used widely for dental surgery, long bone defects, bone nonunion, vertebral fusion operation and maxillofacial repair [76]. The biocompatibility, osseointegration, and bioactivity of metal implants are improved by coating their surfaces with HAP, which enhances the bone contact area and cell adhesion properties of the implants [30]. Moreover, HAP can promote the biomineralization of macromolecule-based scaffolds. HAP nanoparticles penetrate into the demineralized collagen scaffold and serve as mineralization seeds that promote the occurrence of remineralization of the collagen matrix [77].

HAP has high chemical stability, but has weakness in mechanical properties. Stress along the *Z*-axis direction of HAP crystals creates brittleness [78]. It is worth mentioning that wear resistance, the friction coefficient and hardness of dense HAP are similar to natural those of mineralized tissues [32]. The fatigue resistance of dense HAP is superior to porous HAP [64]. As a result, HAP is not used as a load-bearing implant due to its poor mechanical properties, but is usually implanted with granules and porous scaffolding [29]. It is still a huge challenge to improve the mechanical properties of HAP. Metal oxides including zirconia, alumina and titania are common reinforcing agents [79]. However, the biocompatibility and biodegradation properties of HAP-based biomaterials are compromised by the addition of these reinforcing agents, which are bioinert or none-biodegradable [80]. Constructing composites with a polymer is an effective way to improve the mechanical properties of HAP. Natural polymers such as chitosan, hyaluronic acid, silk fibroin and gelatin are common components for fabricating hybrid scaffolds [81]. For instance, the hydroxy propyl methyl cellulose of chitosan has been crosslinked to fabricate chitosan/HAP sponge-like scaffolds which have excellent compressive strength, elasticity and degradability [82]. Considering the importance of mechanical properties for bone repair, particularly in load-bearing bones, further research is necessary to improve the mechanical properties of HAP [83,84].

The biological performance of artificial bone implants is extremely important. HAP is considered to have good biocompatibility and bioactivity in osteoconductivity, but has poor osteoinductivity [75]. Therefore, it is usual to combine HAP with other materials to improve its osteoinductivity. Beta-tricalcium phosphate (β-TCP), another common kind of calcium phosphate used in bone regeneration, has better osteoinductivity than HAP. This biphasic calcium phosphate (BCP) material has been synthesized by combining HAP and β-TCP to take advantage of the properties of both and obtain better bioactivity for bone regeneration [85]. The BCP material possesses superior bioactivity, biodegradability, osteoinductivity, and mechanical properties than HAP or β-TCP alone, and has greater ability to stimulate osteogenic differentiation of BMSCs [86]. Hence, bone grafts and dental materials are commonly prepared with BCP material [87]. Zhu et al. constructed BCP bioceramics with micro-whiskers and a nanoparticle hybrid structure which may be applied in research on load-bearing bone tissue regeneration to provide mechanical support [88].

HAP is an advanced material for preparing bone grafts owing to its similarity to natural minerals and excellent biocompatibility and osteoconductivity. However, the preparation performance regulation of hydroxyapatite materials for bone regeneration remains a long-term and challenging endeavor. First of all, basic research on hydroxyapatite in bone tissue remains largely unexplored. For example, there is a lack of understanding regarding the factors involved in the formation of hydroxyapatite, including precursors and crystal growth regulatory factors, in the process of bone tissue biomineralization. In addition, the interaction between hydroxyapatite and collagen molecules, particularly how the regular hydroxyapatite-collagen complex is formed, needs to be further studied. On the other hand, there are still many technical and scientific problems associated with the synthesis of hydroxyapatite and the preparation of scaffolds. For instance, the mechanism of hydroxyapatite crystal growth needs to be further explored to control the scale, and a controllable fabrication strategy for ordered biomimetic structures must be developed to fabricate scaffolds with good mechanical properties and controllable porous structures.

### 3.3. Tricalcium Phosphate

TCP, as one of the most studied calcium phosphate materials, contains two crystalline phases (α-TCP and β-TCP). There are several phases of CaP materials that have similar compositions to TCP, and the term TCP here is used for the phase with a chemical composition of Ca_3_(PO_4_)_2_ and a Ca/P ratio of 1.5. Pure crystalline α-TCP cannot be precipitated in aqueous solutions since it is very poorly soluble, unlike β-TCP [89,90]. There are three approaches for synthesizing β-TCP, including solid-state reaction, thermal conversion, and precipitation. Usually, crystalline β-TCP is prepared at a high temperature of ~800 °C such as by thermal decomposition of calcium deficient hydroxyapatite (CDHA), and the other is the solid-state interaction between acidic CaP (i.e., DCPA) and alkaline (i.e., CaO) [33]. As well, it has been shown that β-TCP precipitates in organic solutions, such as ethylene glycol, methanol, tetrahydrofurane, and ethyl propionate [91,92,93]. Tang et al. synthesized β-TCP at a relatively low temperature at about 150 °C in organic solvents (e.g., ethylene glycol) [94]. Moreover, β-TCP transforms into the α-TCP at higher temperatures (above 1125 °C), so α-TCP may be considered as the high-temperature phase of β-TCP [38].

TCP has excellent stability and can be stored in a dry environment at room temperature for a long period of time. β-TCP is more stable than α-TCP according to a density functional study [95]. α-TCP has superior reactivity and specific energy in an aqueous solution than β-TCP, and is capable of being hydrolyzed to CDHA [33]. In clinical applications, β-TCP has higher osteoconductivity and osteoinductivity than HAP and is primarily used in bone cements and bioceramics [34,96], while α-TCP is normally used in cements, since it is subject to a phase conversion to HAP upon water contact [97,98]. It should be noted that the rate of resorption of pure α-TCP is higher than new bone formation, which leads an imbalance between the process of bone formation and implant degradation [38]. Therefore, α-TCP is usually used as a component in CaP cements with other CaP materials [33]. In contrast, β-TCP has a relatively lower resorption rate than α-TCP, and has good prospects for application in bone regeneration [32]. The nano-porous structure of β-TCP allows for excellent biomineralization and cell adhesion; these properties can stimulate osteoblast and BMSCs proliferation [99]. In addition, compared to HAP, β-TCP has better biodegradability and resorption rate, which can increase the biocompatibility of the implants for bone regeneration [31].

### 3.4. Amorphous Calcium Phosphates

Amorphous calcium phosphates (ACPs) are a special phase of CaP with various chemical compositions. ACPs have long-range order but short-range disorder regarding their crystal properties [33]. Initially, ACPs were discovered during the preparation of HAP in vitro; therefore, ACPs were considered as precursors of HAP [100]. A study in 1972 found that ACPs were the first phase to form and were transformed into octacalcium phosphate (OCP) during the synthesis of HAP in vitro; the final phase conversion occurred from OCP to HAP [101]. Glimcher et al. believed that ACPs may be the precursor stage of bone formation due to the presence of uniform intra-collagen mineralized particles found in collagen mineralization in vitro through ACPs [102]. ACPs are classified into two groups based on their preparation temperature, namely low-temperature ACPs and high-temperature ACPs [36]. Low-temperature ACPs usually occur as precursors during the precipitation process of other CaP compounds. Since the surface energy of ACPs is lower than that of OCPs and HAPs, ACPs are thought to form at the onset of precipitation [36].

The chemical composition of ACPs depends on pH value and the concentration of calcium and phosphate ions in aqueous solution. The recrystallization of ACPs occurs with increased concentration of Ca^2+^ and PO_4_^3−^. In addition, ACPs may recrystallize slowly or transform into CaP materials with a higher crystalline degree, such as CDHA, in a reaction system with a continuous and mild stirring rate, especially when at higher reaction temperatures [33]. In studies concerning the influence of pH, researchers have discovered that the Ca/P ratio of ACPs increases from 1.18 to 1.53 as the pH value of the system changes from 6.6 to 11.7 [103,104]. Up to now, the atomic distribution in ACPs is still not entirely clear, which is an important research topic in studies of biominerals [36]. Freshly precipitated ACPs usually display spherical-like structures with diameters between 20 and 200 nm as seen by electron microscopy [33]. Some researchers believe that the basic structural unit of ACPs is thought to be a spherical cluster structure with a diameter of 0.95 nm. The ACP chemical formula is Ca_9_(PO_4_)_6_ [36,105].

ACPs constitute the initial phase of HAP and are essential components in the process of bone regeneration and bone mineralization due to their particular physical properties and structure [106]. ACPs possess superior biological properties, such as osteoconductivity and biodegradability, leading to a variety of applications including CaP bone cements, biological tissue engineering scaffolds, bone repair biomaterials, and dental implants [107,108]. In addition, the nano-sized clusters in the ACPs have characteristics of large specific surface areas and pH-responsive degradation, which makes them ideal drug delivery carriers for studies relating to drug loading and controlled release [35].

The preparation of ACPs is regulated by small molecules such as ATP, which can effectively inhibit the phase transformation of ACP [109]. As a result of an ATP-assisted preparation strategy, the product is an ACP composite nanoparticle containing ATP and ADP molecules. Furthermore, the compound has good biocompatibility and osteogenic activity, and can up-regulate the expression of osteogenic genes in BMSCs. An injectable hydrogel prepared by combining ACP compounds with fibrinogen displayed excellent promoting effects in in vivo bone regeneration [110].

An ALP-catalyzed hydrolysis reaction was used to generate EACP nanominerals in an alkalescent aqueous solution similar to mitochondrial surroundings [109]. The mechanism of EACP promoting bone healing was demonstrated in that the ADP/AMP biomolecules and Ca^2+^ ions released from EACP can increase the activation level of AMPK and promote autophagy and osteogenic differentiation in hBMSCs. Additionally, a number of theories suggest that ACP plays an important role in the biomineralization process as a precursor of apatite formation [111,112,113]. There is also evidence that the biomineralization route involves the formation of the mineral phase within matrix vesicles that are associated with small crystals of calcium phosphate mineral [114], which are usually an amorphous phase involved in the formation of these vesicles [115]. Meanwhile, disordered collagen fibrils may contribute to the stabilization of ACP, resulting in both amorphous and crystalline bone mineral [116]. At present, the collagen fibrils as the temptation for bone mineral growth are attracting a great deal of interest in biomineralization research [117]. However, the specific mechanisms by which ACP promotes bone regeneration are highly controversial and require further investigation.

### 3.5. Application of Other CaP Phases

Tetracalcium phosphate (TTCP) is the most basic CaP phase, which is in a metastable state and gradually hydrolyzed to HAP and Ca(OH)_2_ in a humid environment or aqueous solution [38]. TTCP often occurs as a by-product of HAP plasma coating, which is a mixture of α-TCP, TTCP and CaO from the high temperature phase [40]. The chemical synthesis process of TTCP can only be carried out in dry air or vacuum environment. It is synthesized by a solid-phase reaction at over 1300 °C. In the presence of water vapor, TTCP is decomposed into HAP [38]. There are three types of TTCP bone cement: single component, multi component, and polymer. In biological applications, TTCP is usually used as a component for preparing self-curing bone cements, biological composites or root canal sealants [118,119,120]. By combining TTCP with DCPA or DCPD (Ca/P = 2.0), bone cements with the stoichiometry composition of HAP can be produced [40]. A set cement with the best mechanical properties was obtained using an equimolar mixture TTCP and DCPA with a particle size (diameter) ratio of approximately 10: 1 [40]. However, another study found that cement with a diameter ratio (TTCP:DCPA) of 20:1 had the highest compressive strength [121].

TTCP bone cements show advanced biological performance. Qin et al. fabricated three-dimensional porous TTCP scaffolds via selective laser sintering technology (SLS) [122]. After immersion in SBF for one day, nanoapatite was produced on the surface of TTCP scaffolds. The scaffold surface was completely covered with apatite after three days, indicating good biological activity. Furthermore, the biocompatibility of TTCP scaffolds was evaluated by cell culture, which confirmed their high biocompatibility. An evaluation of the histological effects of the TTCP cement applied to the pulp of rat upper incisors demonstrated great advantages over calcium hydroxide (Ca(OH)_2_) cement [123]. Tsai et al. investigated a single component TTCP cement (containing (NH_4_)_2_HPO_4_ as the liquid) in rabbit femurs for 24 weeks in vivo [124]. Following implantation, histological examination of the recovered implants demonstrated good cement-bone host bonding, with cement resorption, new blood vessels, osteoocytes, and osteoblast-like cells identified. At the end of 24 weeks, only a small amount of residual bone cement was found, and the cortical bone was almost completely remodeled.

Octacalcium phosphate (OCP), as a precursor to HAP crystal formation, along with ACP and DCPD, play an important role in bone formation and biomineralization [37,125]. A very similar structure exists between OCP and HAP but OCP is more unstable than HAP and is hydrolyzed to HAP [37]. The mechanism of hydrolysis of OCP is still not completely clear. Two hypotheses, dissolution-reprecipitation mechanism [126] and ion diffusion-crystallization conversion [127], are proposed to explain the hydrolysis of OCP. Eliminating the HPO_4_^2−^ from the OCP water layer has been confirmed as a necessary step for phase transformation, and is believed to be the rate-determining factor of the conversion [128]. The transformation of OCP was observed under in vitro and in vivo conditions. Upon being placed in water with a starting pH of 7.2, the mixture of OCP and HAP was examined after 1 h, and at 12 h the structural transition was completed [129]. The OCP was completely hydrolyzed to CDHA within 6 h in deionized water [130]. pH also affects the transformation rate. For example, Suzuki et al. found that OCP hydrolysis was postponed at pH 11 compared with pH 7.4 [131]. Interestingly, the hydrolysis of OCP into HAP is very slow in in vivo conditions. Implanted OCP in a rat calvarial defect hydrolyzed very slowly after 21 days [132]. In SBF at a temperature of 36.5 °C and a pH of 7.25, the hydrolysis of OCP to HAP took place to a small extent over the course of 28 days [133]. In addition, OCP may be non-stoichiometric, and its structure may be calcium-deficient (Ca/P = 1.26) or calcium-excessive (Ca/P = 1.48) [134].

OCP has good osteoinductivity and is widely used in bone repair research, including the coating of metal grafts, the use of CaP bone cement, and the construction of composite bone repair scaffolds [135,136,137]. OCP/Col composite scaffolds constructed from OCP particles and collagen have a higher osteoconductivity than OCP alone, and the osteoconductivity is positively correlated with the dose of OCP [138,139]. By providing a nuclear structure, OCP acts as an initial deposition site for bone, and its conversion to HAP plays a significant role in bone formation, which may explain its beneficial effects on bone growth [132,140,141]. By implanting the precursors of HAP, such as OCP, ACP, and DCPA, along with HAP particles in the subperiosteal region of the mouse calvaria, bone tissue appeared with OCP in approximately one week. At about 3 weeks, bone tissue appeared in ACP and DCPA, and was later found in HAP particles (5 weeks), which further indicates that OCP has good ability to promote bone formation [140]. Moreover, osteoblasts that can initiate bone formation were found on the surface of OCP particles in the OCP group. Ultrastructural SEM examination confirmed that osteoblasts were directly attached to OCP to form bone matrix, and that filaments were formed around OCP. There were many similarities in the composition of the granulated and granular complexes around with OCP to the bone nodules formed during intramembranous osteogenesis [142].

The application of OCP in bone graft biomaterials and bone regeneration has a promising future due to its good osteoconductivity and osteoinductivity. It is of paramount importance to explore and understand the biological mechanism of good osteoinductivity of OCP, as well as the influence of Ca/P stoichiometry and microstructure on its intrinsic biological activity [143,144].

Dicalcium phosphate anhydrous (DCPA) and dibasic calcium phosphate dihydrate (DCPD) are acidic CaP materials. DCPA is an anhydrous crystalline form of DCPD. Since there are no hydrated molecules, the solubility of DCPA is lower than DCPD. Both can be precipitated from an aqueous solution at 100 °C. The difference between DCPA and DCPD is that DCPA does not form in vivo, as confirmed by no DCPA being formed in normal or pathological calcification nodus [33]. DCPA is often mixed with other calcium phosphate materials to prepare bone cement, and it is also used to provide calcium and phosphorus in foods and toothpastes [38,145,146].

DCPD is the dihydrate crystalline state of DCPA [38]. By adjusting pH in the range of 3–4 at room temperature, DCPD can be produced by the neutralizing reaction of Ca (OH)_2_ and H_3_PO_4_. Metathesis reactions using calcium-containing phosphates in a slightly acidic environment can also synthesize DCPD [145]. Due to its biocompatibility, biodegradability, and osteoconductivity, DCPD is often used as a component of bone cements and toothpaste to promote bone and tooth mineralization [38,147]. It is worth noting that DCPD was found to be converted to calcium deficient hydroxyapatite (CDHA) in vivo [148]. This conversion process released many acidic substances when excessive DCPD was implanted in vivo, causing a severe inflammatory response [39].

Dicalcium phosphate monohydrate (DCPM) as a crystal phase has a Ca/P ratio of 1:1, and is a new metastable CaP with structural water without DCPD and DCPA [59]. DCPM is formed using ethanol and water mixtures that maintain a low level of hydration and inhibit the formation of DCPD. X-ray powder diffraction (XRPD), was used to determine the crystal structure of DCPM, conformed with *a* = 8.0063(4) Å, *b* = 6.7954(5) Å, *c *= 7.7904(5) Å, α = γ = 90°, β = 91.548(4) Å. In addition, after immersion in water for only one hour, this new crystalline form of calcium phosphate monohydrate transforms into hydroxyapatite, which is the stable form of calcium phosphate found in human bones. This represents a two-fold increase in speed as compared to the dicalcium phosphate dihydrate (DCPD) phase, which is usually used in bone cements today. Furthermore, DCPM can be stabilized by organic molecules such as citrate salts, which are abundant in the human body, and can adsorb a large quantity of small molecules. Consequently, DCPM is an interesting option to encapsulate and release drugs to enhance bone healing and remineralization. However, there is still much work to be done on the features and applications of DCPM in bone repair and biomedicine.

And some commercial products of CaPs were displayed in the Table 3.

## 4. Effects of Sizes and Structural Characteristics of CaP Materials

### 4.1. Sizes of CaP Materials

Vertebrates’ bones are composed of multiple levels and sizes of units, ranging from nanometers to micrometers, with precise yet complex arrangements (Figure 2 [149]). In terms of microstructure, the trabeculae thickness ranges between 50 and 300 μm, with their orientation depending on the distribution of load in the bone [150]. Mineralized collagen fibers form lamellae whose width is approximately 3–7 μm [150]. Mineralized collagen fibrils with a diameter about 100 nm are formed on the lamellae at the nanoscale level. These fibrils, 300 nm long and 1.5 nm thick, are the basic building blocks of the bone material and consist of collagen molecules [33]. Crystals grow with an approximate repeat distance of 67 nm on the fibrils [151]. This hierarchal structure of biomineralized fibrils and trabeculae from the nanoscale to the microscale are critical to the isotropic properties in the bone, and enhance load-bearing capability. Therefore, synthesizing different sizes of CaP materials, constructing the biomimetic hierarchal grafts, and understanding the biological performance of various sizes of CaP crystals should improve bone defect treatment.

Over the past decades, many strategies have been developed to control the size of CaP crystals, including chemical precipitation, sol-gel process, microemulsion, a hydrothermal method, solution combustion synthesis and an electrospinning method. In addition, microwave, ultrasonic and gravity precipitation methods, as well as the use of reagents such as proteins, polymers and chelated reagents, regulates the size of CaP crystals. Synthesis methods for controlling the size of CaPs are the initial and key steps to understanding the physiochemical properties and biological performance of different sizes of CaPs. The specific methods and details of these strategies are described in [152]. Several classical and effective strategies are introduced below.

The chemical precipitation approach is the simplest method for preparing CaP crystals. The size of CaP crystals is usually related to the reaction time and temperature [153]. However, the difficulty of precisely obtaining the size of CaP crystals is a big problem using this method. The microemulsion method is highly effective for regulating the size of CaP crystals due to the uniform and narrow channels of the microreactors strictly restricting the nucleation and growth of crystals [154,155]. The surfactants in the microemulsion play a critical role in modulating the CaP crystal size. When the molar ratio of organic solvents to surfactants increases, the length of HAP nanorods increases also [156]. In addition, changing the molar ratio between water and surfactants (W_0_ = H_2_O/surfactant) can also regulate CaP crystal size. HAP products are changed from nanospheres with a diameter about 25–40 nm to the needle-like crystals with 4–8 nm in diameter and 80–100 nm in length when the W_0_ is increased from 5 to 8, and when the increase of W_0_ was from 5 to 15, the HAP products were changed to the rod-like crystals (10–17 nm in diameter and 24–50 nm in length) [157].

The use of chelators is another method for controlling the size of CaPs [152,158]. For instance, HAP nanorods (120 nm in average length and 25 nm in diameter) were obtained in the presence of citric acid, while nanowires (0.7–1 μm in length and 30–40 nm in diameter) were produced in the absence of citric acid (Figure 3) [158]. Regulating the concentrations of precursors can, to a certain degree, tailor the size of CaP crystals via precursor transformation routes. Figure 4 shows that the size and aspect ratio of the β-TCP platelets increases as the concentration of the precursor increases [94,159].

It is known that the properties of CaPs are closely related to their size, which may determine their application potential [42,160,161]. The nanoscale CaP particles used for bioceramics display improved sintering ability in ceramics preparation and prevent microcracks caused by extreme sintering temperature, owing to high surface energy [29]. Implants made from HAP nanoparticles have superior mechanical properties compared to micro-sized HAP crystals [156], and offer an effective way to enhance the mechanical properties of CaP-based grafts. Furthermore, CaP materials with nanoscale crystals have greater absorbability compared with microscale crystals, and are therefore more suitable for use in bone tissue regeneration [156]. In addition, nanoscale CaP crystals are usually used in cell targeting [162], drug/protein/gene delivery systems [163,164] and gene transfection/silencing [165,166].

For biological effects, the size of CaP materials plays a critical role in regulating osteoblast proliferation, cellular activity, apoptosis, and osteogenic differentiation [31,42,161]. And some in vitro/ vivo experiments of CaPs and their markable outcomes were displayed in Table 4. The HAP particles at the nanoscale, particularly those with a diameter of 20 nm, have been shown to promote cell proliferation and bioactivity, and inhibit cell apoptosis compared with those at the microscale [41,167]. These positive effects are probably attributable to the better penetration abilities of smaller HAP nanoparticles (Figure 5) [41]. The size of CaP materials affects bone regeneration by inducing macrophage polarization [168]. HAP crystals at the nanoscale induce M2 macrophage polarization [169], while at the microscale polarize into M1 macrophages [170].
jfb-13-00187-t004_Table 4Table 4In vitro and in vivo experiments of CaPs.First AuthorCaPsIn VitroIn VivoOutcomesReferenceMahon ORHAP nanoparticleshBMSCs, HUVECsRatPromoting M2 macrophages polarization and angiogenesis; Specifically enhancing IL-10 production [171]Ji Cβ-TCP scaffoldrBMSCs, HUVECs, RAW264.7RatSpecifically enhancing the expression of osteoclast differentiation and extracellular space pathway genes to promote the process of bone remodeling[172]Raymond Yα-TCP scaffoldMG-63RabbitHydrothermal process promising a more favorable microstructure, nanoporosity, and nanopore size; significantly enhancing bone formation[173]Zhou ZACP/GelMA scaffoldrBMSCsRatInducting the ALP into the biomimetic strategy to produce mineralized ACP nanoparticles; enhancing the proliferation of BMSCs and upregulating the osteogenic differentiation owing to bioactivity of ALP[174]Kurobane TOCP/gelatin scaffoldHUVECsRatStimulating the angiogenesis then enhancing the bone regeneration. Exploring the relationship between OCP dose and angiogenesis[175]Sheikh ZDCPA cement-RabbitComplete resorption and more bone formation than DCPD cement. Bone formation and resorption in DCPA cement are site specific[176]Ko CLDCPA/DCPD -rich cementmBMSCsRabbitHaving higher cell viability, ALP activity, and ALP quantity. Showing lesser residual implant and higher new bone formation[177]Tsai CHTTCP cement-RabbitThe new bone formed started at the center of TTCP cement at 12 weeks. The resorption of grafts, bone ingrowth and remodeling activities were completed at 24 weeks.[124]hBMSCs: human bone mesenchymal stem cells; rBMSCs: rat bone mesenchymal stem cells; mBMSCs: mice bone mesenchymal stem cells; HUVECs: Human Umbilical Vein Endothelial Cells.


It is noteworthy that the component sizes, the surface and pore scale levels of the CaP grafts and hybrid scaffolds are closely related and are important to bone regeneration, considering that the grafts and scaffolds are the main forms of bone implants. Li et al. demonstrated that compared to CaP bioceramics fabricated by microcrystalline structures, bioceramics using nanocrystalline structures have many advantages in the form of unique surface topography, good bioactivity, excellent osteoinductivity and proper biodegradability. Accordingly, the CaP nanoceramics have significantly superior biological performance in promoting osteogenic differentiation and bone formation (Figure 6) [161,178].

### 4.2. Pore and Surface Characteristics of CaP

Another key factor influencing the biological response of cells and the effects of bone repair is the pore size of scaffolds. Macropore and micropore structures exist in CaP ceramics or scaffolds. The macropore structure is designed to promote osteoinduction [42]. Numerous studies have confirmed the positive effects of macropores on cellular growth and tissue formation [179,180,181]. Macropore sizes in range of 300–500 μm are recommended because this scale ensures the transportation of nutrients and metabolites, osteogenesis and vascular ingrowth [42,182]. Macropores ranging from 300 to 500 μm may possess optimal surface tension, which has been demonstrated as an important determinant of cell adhesion and tissue growth, to the mechanoreceptors of cells [183].

Additionally, micropore structure, generally characterized by pores smaller than 50 μm, is critical for the CaP scaffolds. The micropore structure is thought not only to improve bone ingrowth, but also to create additional space for bone formation [184,185]. Furthermore, a larger surface area, attributed to the presence of micropores, can enhance protein adsorption, ion exchange and mineralization [186]. Moreover, bone tissue regeneration is accelerated due to the capillary effect induced by micropores, which further improves the homogeneity of bone distribution in the scaffolds [187]. Interestingly, some studies have found that new bone tissue formation was observed in HAP ceramics with micropore structures that were subcutaneously and intramuscularly implanted in dogs, and did not occur in ceramics without micropores [184,188]. Micropore structure is becoming increasingly important as research advances. Bohner et al. found mineralized cell/collage-rich tissues only in scaffolds with micropores having threshold sizes greater than 1–10 μm, and were not related to macropore size [185]. Regulating micropore size from 1.58 ± 0.65 μm to 0.65 ± 0.25 μm in TCP ceramics resulted in more abundant bone tissue, which further confirmed the importance of micropore structure [189].

The surface structures of scaffolds or ceramics at the microscale to nanoscale can influence cell adhesion, spreading, cytoskeletal distribution, and gene expression (Figure 7) [42,190]. Holthaus et al. [191] found that microgrooves with 60–100 μm widths had more cells (35–45%) growing than those of with widths of 20–40 μm (16–25%). In contrast, the opposite phenomenon was observed with respect to cell orientation. The number of aligned cells along the microgrooves was higher at 20–40 μm (64–79%) widths than at widths of 60–100 μm (29–47%). According to the study, the size of microgrooves similar to cell dimensions may be beneficial in guiding cell alignment and adhesion. Notably, the sensitivity of different cells to microstructure sizes varies. For example, microgrooves with 8 μm widths responded strongly for cell alignment of both myoblasts and osteoblasts, while those with the widths of 24 μm only affected myoblasts [192]. The depth of the microgrooves influences cell adhesion by single cell guidance. The adhesion force of an individual osteoblast cell to the matrix decreases with increasing depth of the microgroove from 3 μm to 5.5 μm [193].

Nanoscale surface structures of scaffolds and ceramics have been widely applied and have received much attention since the development of nanofabrication techniques [194,195]. Surface structures of scaffolds/ceramics on the nanoscale may affect the cells’ response due to the ECM being composed of nanosized collagen fibrils and cellular receptors and filopodia which are also at the nanoscale [42]. In addition, many studies have been conducted on mechanisms responsible for osteogenic differentiation of BMSCs in response to surface structures and sizes [196,197,198]. However, more studies are needed to understand how nanoscale surface structures at different levels influence cell behavior and responses. The specific relationships between the mechanisms and the surface scale are not clear but have been discussed extensively in [42].

## 5. Regulating Morphologies of CaP Materials

There is a wide variety of shapes and sizes among the natural CaPs. During bone mineralization in vivo, the average crystal size of minerals is smallest at the beginning of formation, and then gradually grows with maturity, leading to various sizes and shapes of crystals [199]. In the past, CaPs with nanoparticle shapes were believed to have better properties. However, minerals with needle-shaped and rod-shaped crystals were found in bone mineralized collagen [200], leading to a controversy regarding the effect of mineral crystal morphology on its properties. Sphere-shaped HAP nanoparticles are beneficial for the proliferation and migration of osteoblast compared to the rod-shaped nanoparticles [201]. Therefore, the morphologies of CaPs play an important role in biological responses and bone tissue regeneration.

With the development of preparation techniques and improved understanding of the mechanism of CaP formation, various CaPs with different morphologies can be produced in vitro by controlling the conditions. The commonly used preparation methods of CaP materials include co-precipitation, emulsion, hydrothermal, microwave-assisted, hydrolysis, solution combustion synthesis, and the sol-gel method [202,203,204,205,206,207]. CaPs with various morphologies can be prepared by adjusting pH, temperature, organic additives, the Ca/P ratio, and ion saturation of the reaction system [149,208].

pH affects precipitation and crystallization of HAPs by influencing the balance between the hydrogen-containing anions and orthophosphates [152]. Generally, based on temperature, they can be divided into two categories: high temperatures (~800 °C) and low temperatures. For example, TCP is required to be synthesized in a high temperature environment, but this high temperature solid-phase reaction method cannot produce uniform CaP nanoparticles [149]. Figure 8 displays the various shapes of β-TCP particles which have been prepared at different pH values and temperatures [159].

Organic additives, such as, hexadecyl trimethyl ammonium bromide [209], poly (acrylic acid) [210] and allylamine hydrochloride [211], affect the morphology of CaP materials. Organic additives regulate the morphology of CaP by the electrostatic interaction between the surface of the crystals and additives, and by regulating the zeta potential of crystals [212,213]. For instance, large HAP nanoparticles with high length-diameter ratios are obtained when adding poly(L-lysine), while using poly (L-glutamic acid) with higher charges results in the HAP nanoparticles with a smaller size [214]. On the other hand, the Ca/P ratio and reaction time are crucial factors in determining the morphology of CaP materials [208,215]. By varying the Ca/P ratio and reaction time, Zhang et al. prepared HAP microtubes and HAP nanowires [216].

Throughout the past decade, CaPs have been reported with a range of morphologies, including particles, spheres, rods, needles, wires, sheets, flakes, strips, porous structures, and hollow structures, all with different sizes ranging from the nanoscale to macroscopical [152,217]. These shapes can be classified into four groups according to dimensions [152]; Figure 9 shows different morphologies of CaPs.

Clusters, particles, and quantum dots are typically zero-dimensional (0-D) structured materials. Zero-dimensional-shaped CaPs have wide applications in delivery systems and the fabrication of CaP bioceramics and composites [235,236,237,238]. One-dimensional (1-D) CaP crystals represent structures with a length that is significantly greater than the cross-sectional dimension, e.g., needles, rods, wires and fiber-like structures. Two-dimensional (2-D) shaped CaP crystals contain sheets, disks, flakes and platelets, which have excellent molecular adsorption abilities and mechanical properties. It is well known that the physical properties of inorganic components, including size and morphology, affect the mechanical properties of the organic-inorganic composites [239,240]. As a result, 1-D and 2-D-shaped CaP powders are commonly used as raw components for improving the mechanical properties of bio-composites, where the latter are usually considered to be the most effective stiffeners in isotopic composites [241].

In recent years, three-dimensional (3-D) CaP materials have attracted a great deal of attention owing to their superior biological performance and spectrum of biomedical applications [152]. Novel 3-D CaP architectures reported in the literature contain porous and mesoporous spheres, hollow and tube structures, ordered and patterned arrays, and porous scaffolds [242,243,244,245]. To fabricate 3-D CaPs, self-assembly and biomineralization methods are employed. These strategies use nanoparticles, nanorods, nanobelts, and nanosheets as building blocks to construct three-dimensional materials with various morphologies [246,247,248]. Furthermore, amino acids, proteins, and surfactants are commonly utilized and are most effective assistance in the technology of self-assembly and biomineralization for controlling the morphology of 3-D CaP functional materials [152].

## 6. Biomimetic Calcium Phosphate Scaffolds for Bone Regeneration

Highly organized arrays of HAP crystallites on the nanoscale level, and intricate bundles of aligned crystallites on the microscale level, are found in human bones and teeth, and are critical in determining many of the advanced biological and mechanical properties of bones and teeth [249,250]. Therefore, fabricating CaP biomaterials that mimic the structures of bones and teeth is a new strategy to improve the performance of biomaterials. Three-dimensional grafts and scaffolds with nano-/microstructured surfaces exhibited better biological properties owing to their similarity to human bones and teeth, and provided promotion of osteointegration and subsequent bone tissue regeneration [251,252,253].

Traditional methods to fabricate 3-D grafts with nano-/microstructured surfaces are assisted by organic solvents and reagents for directing structure. HAP columnar structures were formed elongated in the *c*-axis on the surfaces of HAP bioceramics and on the substrates of metals by a molecular template directing biomineralization method in a SBF system (Figure 10A–C) [254,255]. Active groups and components such as PO_4_^3−^, -NH_2_, -COOH and polydopamine have been developed to promote the biomineralization of CaP crystals because of their excellent ability of capturingCa^2+^ ions, leading to the formation of nuclei that can induce CaP crystal growth [244,256].

Mineralization via SBF soaking usually results in low crystallinity [257]. Large amounts of surfactant and additives, some of which are hazardous to health and the environment, are need to be added to SBF to assist mineralization [258]. In recent years, 3-D printing technology has allowed fabrication of 3-D grafts with nano-/microstructured surfaces [253,259]. Wu et al. [260] mimicked the nano-/microstructural hierarchy of natural wood to fabricate biomimetic hierarchical porous scaffolds by 3D printing technology (Figure 11). It was found that the first-level macropores of the biomimetic, natural, wood-like, hierarchical structure scaffolds had good performance in promoting bone tissue ingrowth, whereas the second-level micro/nanoscale pores performed well in transporting nutrients and metabolites. These scaffolds with nano-/microstructured surfaces exhibited excellent features in osteoinductivity and bone tissue regeneration.

Microporous CaPs scaffolds with nano-/microstructured surfaces have an excellent capacity to transport cells, resulting in a high interest in the field of bone tissue engineering [261]. Three-dimensional microporous HAP bioceramic scaffolds have been constructed via hydrothermal method by using α-TCP bioceramic scaffolds as precursors without any assistance from structure-directing reagents and organic solvents [261]. The 3-D ordered scaffolds have highly interconnective macropores and various surface topographies such as nanosheets, nanorods and microrods that can be tailored by regulating the reaction medium of the NaH_2_PO_4_, Na_3_PO_4_ and CaCl_2_ aqueous solution (Figure 12) [251].

And the HAP bioceramics with biomimetic nano-/microstructured hybrid scaffolds demonstrated excellent adhesion, proliferation, and osteogenic differentiation capabilities of BMSCs. In addition, excellent protein adsorption due to their surface characteristics contributed to the bioactive performance of the composite scaffolds, which resulted in enhanced bone regeneration. Furthermore, the mechanism of the biomimetic hybrid scaffolds with nano-/microstructured surfaces was investigated. It was found that integrins were activated initially by the attachment of cells to scaffolds, and that the BMP2 signaling pathway and related Cx43-based cell-cell interactions were activated subsequently, in addition to an interaction between BMP2 and Cx43 that facilitated osteogenic differentiation (Figure 13) [198].

## 7. Drug Delivery Applications and the Potential of CaPs

CaPs, especially porous nanostructured CaP materials, are widely used as drug delivery carriers due to their suitable architecture, large surface area and stability in biological fluids. As well-known, CaPs dissolve at a slightly acidic pH, which makes controlled delivery of drugs into cells possible [262]. Furthermore, the production of the ionic, non-toxic constituents Ca^2+^ and PO_4_^3−^ after dissolution of CaPs can prevent particle accumulation and induce the release of drugs into the cells [263]. As a result, CaPs are commonly used as carriers for the delivery of commercial drugs, bioactive molecules and genetic materials [264,265,266].

The drug delivery behavior of CaPs is influenced by characteristics including crystallinity, microstructural properties, surface area and charge, particle size, and morphology [267]. Higher levels of drug complex loading were observed for HAP nanoparticles with lower crystallinity and higher surface area when compared to similar nanoparticles with higher crystallinity [268]. Lower release rates were observed with CaP particles of higher crystallinity, whose solubility is affected by their crystallinity, resulting in a decrease in drug release rate [269]. Additionally, the morphology of the CaPs results in different surface areas, resulting in different drug loading efficiency. In a study by Palazzo et al., therapeutic drugs were absorbed onto plate-like HAP particles approximately 1.3 times more than onto needle-like particles [270]. In addition, spherical CaP nanoparticles provide more effective drug loading and release properties than particles with flaky, brick-like, or elongated orthogonal morphologies [271]. In terms of particle size, particles with a size of 20 ± 5 nm are best accepted by osteoblast cell lines [272]. Hence, the CaP nanoparticles are favored in the drug delivery system.

A number of therapeutic factors have been delivered by CaP nanoparticles, including antibiotics, anti-inflammatory drugs, and growth factors for bone healing [267]. A CaPs-based antibiotics delivery system is mainly used for the treatment of bone defects that are infected or caused by infection. Moreover, by following this approach, high concentrations of antibiotics are only found at the anatomical sites of interest, thereby minimizing the toxic effects of antibiotics [273]. In addition, growth factors, including the family of BMPs, transforming growth factor-beta (TGF-β), platelet-derived growth factor (PDGF) and vascular endothelial growth factor (VEGF), are critical in the processes of bone repair [274]. Hence, the addition of growth factors can further enhance the osteogenic ability of CaPs. BMP-2 interacts with HAP via the functional groups -OH, -NH2, and -COO [275]. The continuous releasing period can be delayed to 15 days in vitro when delivered by HAP nanoparticles [276]. It is important to note that when these growth factors are loaded onto CaPs, attention must be paid to preventing denaturation of the protein, reducing its functionality.

Gene delivery participates in the promotion and facilitation of bone regeneration [277]. Furthermore, a new approach to tissue regeneration is being brought about by the most recent discoveries and advancements in gene delivery [267]. This makes calcium phosphate nanoparticles an attractive option for bone regeneration.

## 8. The Mechanism of Calcium Phosphate Promoting Osteogenesis

CaPs have applied to bone tissue regeneration engineering for decades, and show advanced osteoconductivity, osteoinductivity and bone healing effects. However, the mechanisms of CaPs in promoting bone regeneration is still a mystery. Many mechanisms have been proposed and confirmed such as osteogenesis, vascularization, neuralization, inflammatory and immunology; some classical mechanisms are reviewed in this work.

### 8.1. Osteogenic Differentiation

The mobilization and recruitment of BMSCs from the bone marrow through the peripheral circulation play an important role in the repair of bone defects [278]. At the cellular level, bone tissue regeneration is determined by the induction and promotion of directed differentiation of BMSCs into osteoblasts following recruitment to the site of the bone defect. With the degradation of the CaP material, Ca^2+^ and PO_4_^3−^ are gradually released, leading to a local ion concentration above the physiological level. As a result of the increased ion concentration, osteoblast proliferation and differentiation and the subsequent process of bone formation are significantly affected [279]. As well, Ca^2+^ is an important homing signal that facilitates multicellular processes such as bone remodeling and wound healing by bringing together different types of cells [280]. A high calcium concentration, for example, has been shown to stimulate the migration of BMSCs [281], pre-osteoblasts [282], and osteoblasts [283] to the site of bone resorption and their maturation into new bone-producing cells. Moreover, the influx of Ca^2+^ increases intracellular Ca^2+^ levels, thereby increasing the polarization of the membrane front, which is crucial for determining the direction of sustained cell migration [284,285].

Apart from this, extracellular Ca^2+^ plays an important role in maintaining osteoblast proliferation and differentiation near the site of bone resorption via calcium/calmodulin signaling [286]. Addition of elevated Ca^2+^ levels to osteoblastic cell cultures has been shown to have an effect on bone cell fate independently of systemic calciotropic factors in a concentration-dependent manner [287,288]. The presence of Ca^2+^ also stimulates the expression of osteogenic growth factors, including parathyroid hormone-related peptide [289], BMP-2, and BMP-4 [290]. In addition, implants enriched with Ca^2+^ significantly improve in vivo osseointegration and bone formation; for example, in a CaP glass, titanium substrate implanted with Ca^2+^, a collagen gel exposed to Ca^2+^, and CaP-coated implants [291,292,293]. In an encouraging development, Ca^2+^ has recently been implicated as an important messenger involved in the non-canonical Wnt/calcium signaling pathway for bone formation independent of the β-catenin pathway [294]. In this pathway, calcium-sensitive enzymes such as Ca^2+^-CaMKII, protein kinase C, and calcineurin are activated by an intracellular release of calcium.

In addition, PO_4_^3−^ also contributes to the proliferation and differentiation of osteoblasts. Several osteogenesis-related biological processes are affected by PO_4_^3−^ as an important signaling molecule, such as osteogenic associated gene expression (e.g., osteopontin) [295] and bone-related protein secretion (e.g., matrix Gla protein (MGP)) [56]. Furthermore, inorganic phosphate (Pi) has been shown to play a crucial role in the physiological mineralization of bone matrix, which is mediated by the enzyme ALP [296]. As well, BMP-2 stimulates Pi transport by osteogenic cells primarily through sodium-dependent phosphate transporters to induce bone matrix calcification [297].

It should be noted that the structures of CaP biomaterials also activate biological signals associated with adhesion, proliferation, and differentiation of cells, such as dimensions, geometries, porosity/microporosity, grain size, and surface topography. Cell sensor and adhesion-related signal pathways play a primary and key role in this biological process. The ERK and p38 MAPK signaling pathways became phosphorylated when BMSCs are cultured on micro/nano structured HAP scaffolds, and when the signaling pathways are blocked by inhibitors, osteogenic differentiation is attenuated [251]. In addition, the MAPK/ERK signaling pathway has been shown to play a significant role in the regulation of cell functions such as proliferation and differentiation [197]. Consequently, BMSCs may have sensed micro/nanostructured surfaces through focal adhesion formation and subsequently activated ERK and p38 signaling pathways, resulting in the upregulation of relevant genes and the formation of osteoblasts. In addition, the TGF-β/BMP and Wnt signal axis may contribute to the structure-sensing of BMSCs and the subsequent osteogenesis as shown in Figure 7 [42].

### 8.2. Vascularization

For bone regeneration to be successful, adequate and rapid vascularization is necessary. In addition, vascularization plays an important role in the viability of seeded cells within the CaP implants. many studies have found that materials can influence the angiogenesis of CaP in bone formation.

Firstly, Ca^2+^ plays an important role in angiogenesis by mediating the angiogenesis-related signaling pathway [298]. In addition, extracellular calcium has been suggested to be a key factor in bone marrow progenitor cell angiogenic responses [299]. Furthermore, the physical characteristics of the CaP material, such as porosity and pore size, are known to influence vascularization [300]. The size of the channel influences the behavior of vascularization in vivo. A large porosity (>50 μm) is necessary for cellular infiltration of bone ingrowth [301]. In a study [300] examining the relationship between pore size and angiogenesis, a scaffold with a pore size of 150 μm displayed significantly greater vascularization than scaffolds with 100 μm and 120 μm pores. In addition, the authors demonstrated that scaffolds with a pore size of 150 μm enhanced the formation of new blood vessels through the PI3K/Akt pathway. The expression of the representative angiogenic factors HIF1α, PLGF, and the migration factor CXCR4 were increased in pores with a 250 μm diameter, and by increasing the diameter of the pores to 500 μm, VEGF expression was enhanced, which led to the development of large vessels [302].

On the other hand, it is known that the CaP can accelerate the BMP2 expression of BMSCs [303], which may be due to Ca ions subsequently activating the PKC, EK1/2, and ERK1/2 pathways, and then entering the nucleus to up-regulate BMP-2 expression through Fos expression and activator protein 1 (AP-1) formation [304]. As well as regulating osteogenesis, BMPs also play an important role in regulating angiogenesis [305]. Moreover, endothelial cells and pericytes are responsible for angiogenesis, and they may also be able to differentiate into osteoblasts when inflammation is present [303]. CaP biomaterials have recruiting ability for these cells, as demonstrated previously [282]. Furthermore, these cells are capable of expressing cytokines such as BMP-2 and BMP-7, which are markedly up-regulated in response to inflammatory stresses [306,307]. In this regard, angiogenesis induced by CaP materials may affect osteogenesis by a variety of mechanisms. One is by secreting cytokines, such as BMPs, and MSCs are recruited to undergo osteoblastic differentiation, and by providing more endothelial cells and pericytes to transform into MSCs, which are then induced to differentiate into osteoblasts under the influence of different materials or cytokines [303].

## 9. Conclusions and Outlook

In summary, calcium phosphate-based materials are commonly and widely used for bone defect repair based on their unique properties resulting from similarity with the main inorganic components of human bones and teeth. Numerous studies have been conducted and have confirmed the excellent characteristics of their physicochemical properties and biological properties, including mechanical properties, biodegradability, biocompatibility, bioactivity, osteoinductivity and osteoconductivity. With preparation technologies developing, the number of CaP materials with varying species, sizes, and morphologies has mushroomed. Different CaP materials have different properties, which determine the approaches of application in accelerating bone tissue regeneration. The CaP materials with different species, sizes and structures have various physico-chemical and biological properties, including mechanical properties, specific surface areas, roughness, porosity, bioactivity, biodegradability, and osteoinductivity. To develop better bone repair materials, it is critical to understand these properties and the biological effects of different species, sizes, and structures of CaP materials. Applying bone repair materials in different approaches in accordance with their advantages is effective for bone regeneration. However, the mechanisms of controlling accurately the species, sizes, and morphologies, as well as the specific relationships between the species, sizes, and shapes of the CaP materials and their biological responses, are not well understood currently. Therefore, determining these relationships will be of great assistance in fabricating artificial CaP bone grafts and in repairing bone defects.

Considering the existing knowledge and available limitations, future studies may focus on the following areas.

(1) CaP materials have a huge potential for bone repair application because of their unique properties similarity to natural minerals. Various CaP materials and composites have been studied and developed for bone regeneration using the advanced physiological properties of CaP such as biocompatibility, osteogenic and osteoinductivity. However, clinical orthopaedic products of CaPs are still rare. With better understanding of biomimetic principles, biomimetic CaP materials have received more attentions. Exploration of the mechanism of biomineralization is important and beneficial for the development and application of CaP-based biomaterials.

(2) Many strategies have been investigated for fabricating CaP materials with different sizes and morphologies. However, strategies for precisely controlling the size of CaPs are still limited. The microfluidic system and microemulsion technologies are novel approaches which may be useful in regulating CaP material sizes accurately.

(3) Many studies have investigated the microscale of the scaffolds/ceramic surface structure and how this affects cell behavior, and have suggested that sizes of the surface microstructure with similarity to cell dimensions might induce the guiding of cell alignment and adhesion. In the future, biological performance and mechanisms of cells responding to different surfaces at the nanoscale will be of interest to researchers.

(4) By understanding natural bone structure thoroughly, artificial bone grafts with biomimetic hierarchical nano-/microstructures have garnered much attention, with promising results for the repair of bone defects. However, biomimetic construction with properly oriented structures ranging from the nanoscale to the microscale remains a challenge. Development of new technologies of mineralization, such as utilizing novel directing biomolecules and precursors, 3D printing with a more precise printing systems and exploiting new bio-inks may be promising routes to achieve these goals.

(5) Up to now, the mechanisms by which the CaP grafts influence bone regeneration are not well understood, such as the influence of nano-/microstructured surfaces on cell behavior. Thus, there is still much work to be done on the mechanisms and relationships between the sizes/shapes of CaP materials and their biological abilities. Further research will result in beneficial developments for the design of CaP biomaterials and for accelerating bone regeneration.

(6) Strategies in improving the physicochemical properties and biological performance of CaP materials are still emerging. For instance, HAP has good biocompatibility but poor mechanical properties. The design and preparation of hybrid scaffolds of CaPs in combination with organic polymers are effective and simple methods. In addition, metal ions as additives can improve the biological activities of CaP-based biomaterials. The addition of Mg^2+^ into CaPs has been shown as an effective strategy to induce the proliferation of osteoblasts and inhibit the absorption of osteoclasts. Meanwhile, other ions, such as Cu^2+^, Fe^3+^, Ti^2+^, and Mn^2+^, may produce unexpected biological responses, which will require further study.

## Figures and Tables

**Figure 1 jfb-13-00187-f001:**
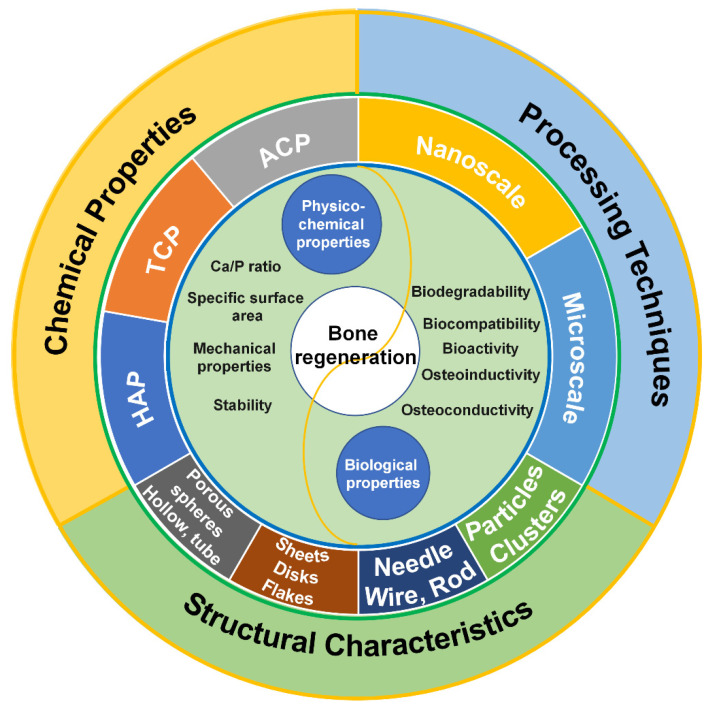
Factors affecting calcium phosphate-based biomaterials for bone regeneration, including chemical properties, processing techniques and structural characteristics.

**Figure 2 jfb-13-00187-f002:**
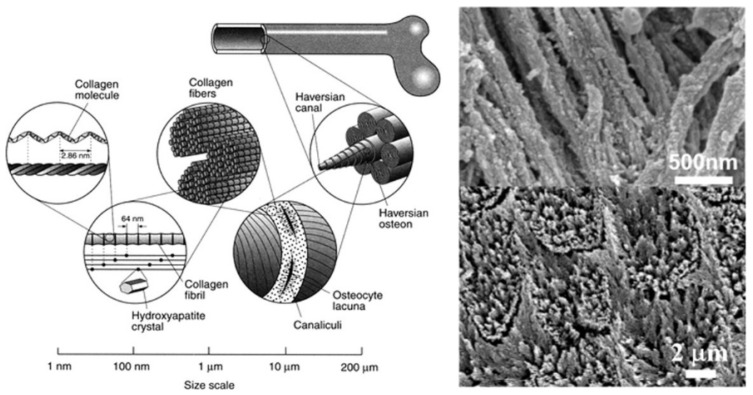
Bone (left) is a complex, hierarchically structured biological material in which the building components are precisely arranged at scales spanning six of orders of magnitude. The image on the left shows sketches of the structural elements of cortical/compact bone (which comprises the harder, outer layer of the cross-section of bone, surrounding the softer trabecular/spongy/cancellous bone) at different scales. The image on the upper right side shows the nanostructure of mineralized collagen fibers in bone. HAP particles are incorporated within the organic matrix. The image on the below right displays the fine structure of dental enamel, the hardest substance in the body, which is composed of an almost pure mineral with elongated HAP nanofibers connected into bundles and forming equally uniaxially directed enamel rods. Reprinted with permission from Ref. [149]. 2011, Uskokovic, V.

**Figure 3 jfb-13-00187-f003:**
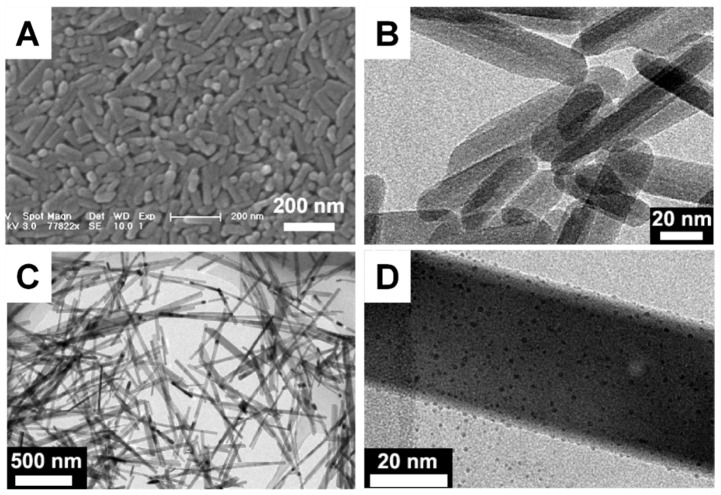
SEM and TEM images of HAP crystals with different sizes after in the addition (**A**,**B**) and in the absence of citric acid (**C**,**D**). Reprinted with permission from Ref. [158]. 2019, Zhang, C. M.

**Figure 4 jfb-13-00187-f004:**
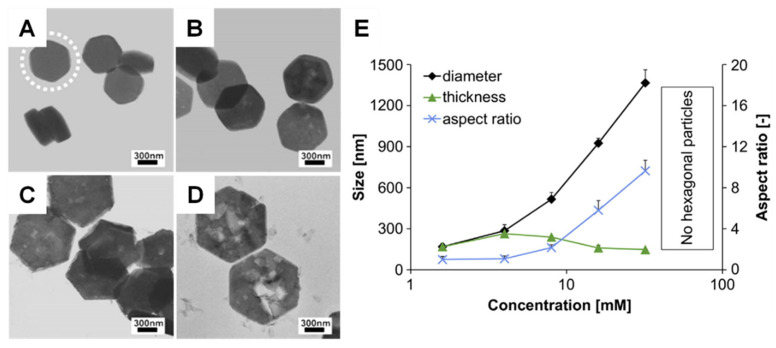
TEM images of β-TCP platelets with different precursor amounts of (**A**–**D**) 10, 15, 26 and 40 mg, respectively. (**E**) Effect of precursor levels on the dimensions of hexagonal β-TCP platelets. Reprinted with permission from Ref. [94]. 2009, Tao, J. H. and Ref. [159]. 2013, Galea, L.

**Figure 5 jfb-13-00187-f005:**
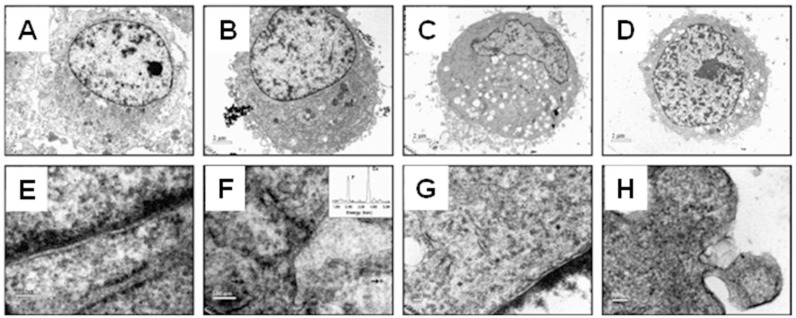
TEM images of MG-63 cells incubated with glass: (**A**,**E**) HAP particles at 20 nm (**B**,**F**), 80 nm (**C**,**G**) and the microscale (**D**,**H**). The black arrows in F indicate 20 nm HAP particles endocytosed by the cell. Reprinted with permission from Ref. [41]. 2009, Shi, Z. L.

**Figure 6 jfb-13-00187-f006:**
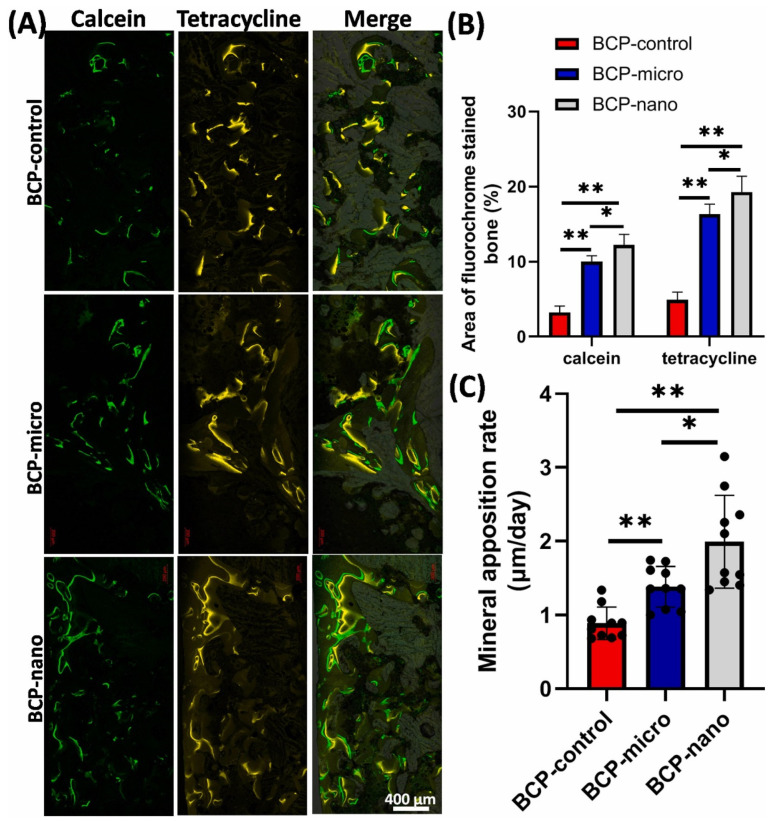
Fluorochrome-labeling analysis of new bone formation and mineralization with different BCP bioceramics at the microscale (BCP-control: irregular shape; BCP-micro: spherical shape) to nanoscale (BCP-nano: spherical shape) grain size. And (**A**) displayed the calcein-labeled newly formed bone at week 8 (column 1, green), tetracycline at week 10 (column 2, yellow), and merged images of the two fluorochromes (row 3). (**B**,**C**) analysis of the fluorochrome-labeled new bone area and mineral apposition rate. Reprinted with permission from Ref. [161]. 2022, Li, X.

**Figure 7 jfb-13-00187-f007:**
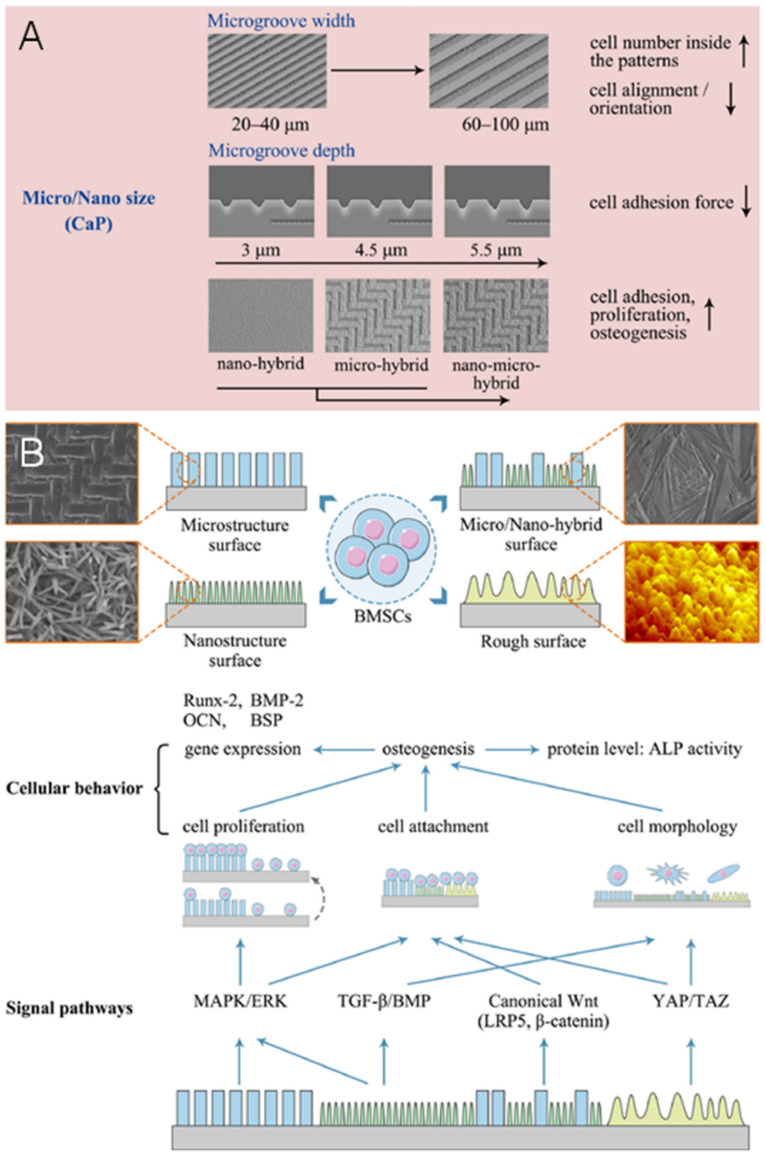
(**A**) Biological effects of surface structure sizes of grafts and scaffolds on cell behavior. (**B**) Different signaling pathways participated in osteogenic differentiation of BMSCs responding to various surface structures. Reprinted with permission from Ref. [42]. 2020, Xiao, D. Q.

**Figure 8 jfb-13-00187-f008:**
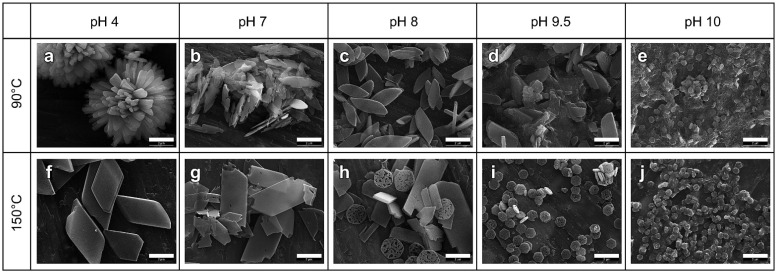
SEM images of β-TCP particles obtained at 90 °C (**a**–**e**) and 150 °C (**f**–**j**), and pH 4 (**a**,**f**), pH 7 (**b**,**g**), pH 8 (**c**,**h**), pH 9.5 (**d**,**i**) and pH 10 (**e**,**j**). Scale bar is 2 μm on all images. Reprinted with permission from Ref. [159]. 2013, Galea, L.

**Figure 9 jfb-13-00187-f009:**
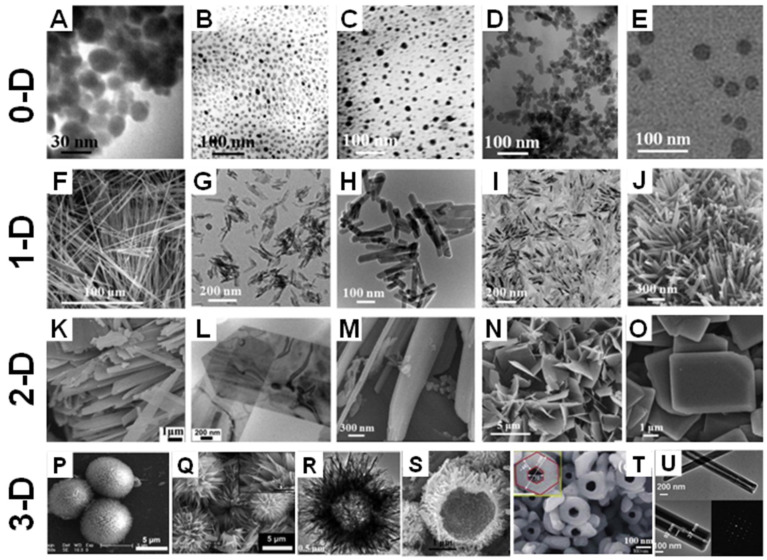
Different morphologies of CaP materials from 0-D to 3-D. (**A**–**E**), 0-D CaP crystals prepared by a milling method (**A**) [218], sol-gel method (**B**,**C**) [219], and microemulsion method (**D**,**E**) [220,221]. (**F**–**J**) 1-D shaped CaP crystals show HAP whiskers (**F**) [222], dicalcium phosphate dihydrate (DCPD) whiskers (**G**) [223], HAP nanorods (**H**,**I**) [224,225] and HAP nanowires (**J**) [226]. (**K**–**O**) displays 2-D shaped CaP crystals including HAP (**K**–**N**) [226,227,228,229] and DCPA (**O**) [230] nanosheets. HAP microspheres (**P**) [158] and flowers (**Q**) [158] obtained by self-assembly with nanorods and microsheets, respectively. (**R**,**S**) [231,232] Typical images of hollow HAP microspheres. (**T**,**U**) [233,234] HAP nanotubes.

**Figure 10 jfb-13-00187-f010:**
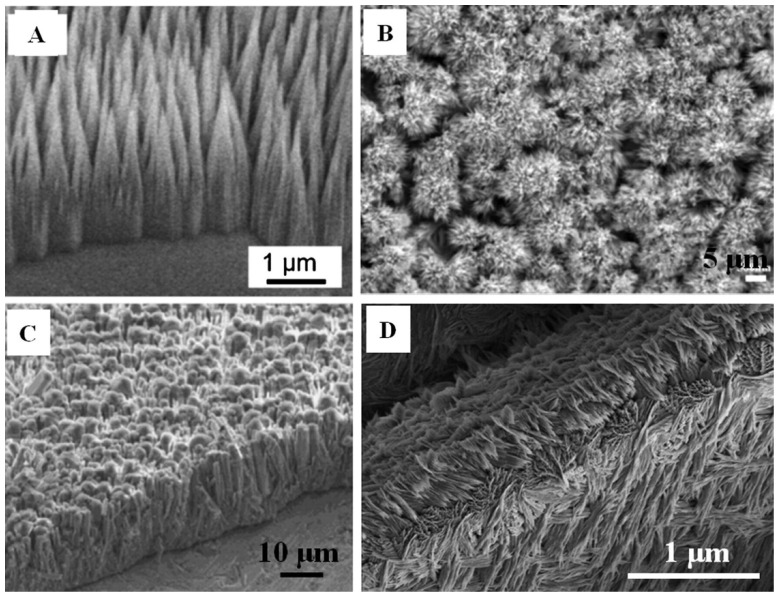
SEM images of bone grafts with 3-D architecture nano-/microstructured surfaces. (**A**) HAP columnar elongated along the c-axis on the HAP disks induced by aspartic acid in SBF; (**B**) highly packed and aligned FHAp coating with enamel-like structure on Ti plates via biomimetic growth in SBF; (**C**) bioinspired crystallization of continuous HAP films on titanium surface induced by Langmuir monolayer of zein protein; (**D**) multilevel hierarchically ordered artificial biomineral HAP ceramic with macroscopical size more than 1 cm using DCPD precursor transformation method. Reprinted with permission from Ref. [152]. 2014, Lin, K.

**Figure 11 jfb-13-00187-f011:**
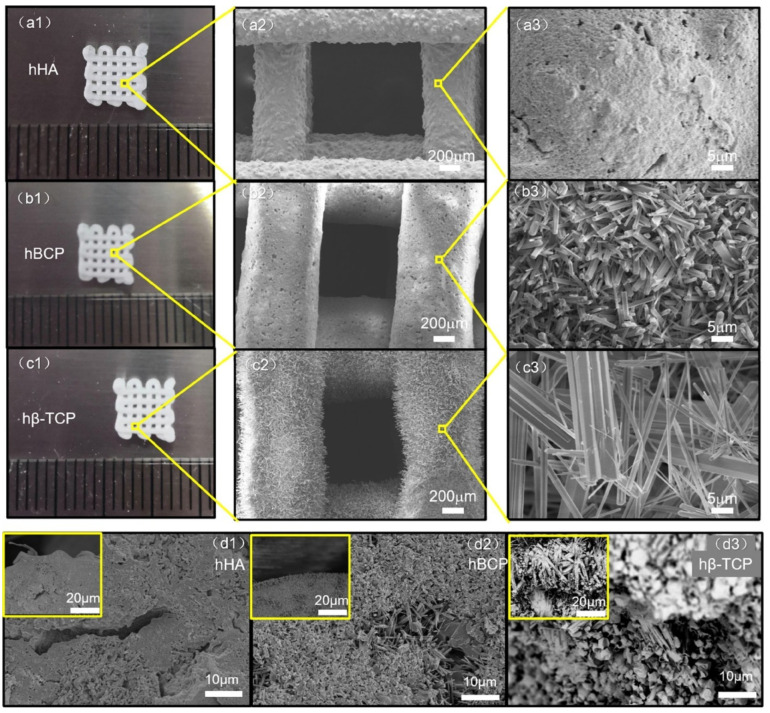
SEM images of 3D-printed biomimetic hierarchical porous calcium phosphate bioceramic scaffolds (hHA, hBCP and hβ-TCP). External macroscopic and microscopic morphologies of hHA (**a1**–**a3**), hBCP (**b1**–**b3**), and hβ-TCP (**c1**–**c3**). Internal microscopic morphologies of hHA (**d1**), hBCP (**d2**), and hβ-TCP (**d3**). Smaller images in the upper left of (**d1**–**d3**) showing scaffold cross-sections of hHA, hBCP, and hβ-TCP, respectively. Reprinted with permission from Ref. [260]. 2020, Wu, L.

**Figure 12 jfb-13-00187-f012:**
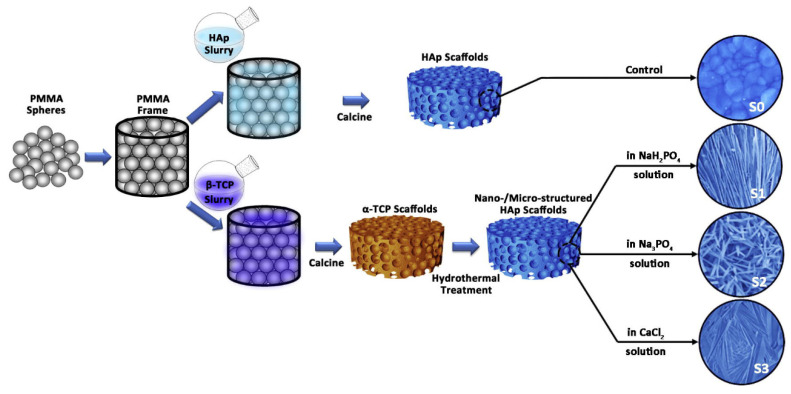
The schematic of the fabrication process for HAp bioceramic scaffolds with different surface topographies: (S0) smooth and flat surface, (S1) nanosheet surface, (S2) nanorod surface, (S3) micro-nano-hybrid surface. Reprinted with permission from Ref. [251]. 2013, Xia, L. G. and Ref. [152]. 2014, Lin, K.

**Figure 13 jfb-13-00187-f013:**
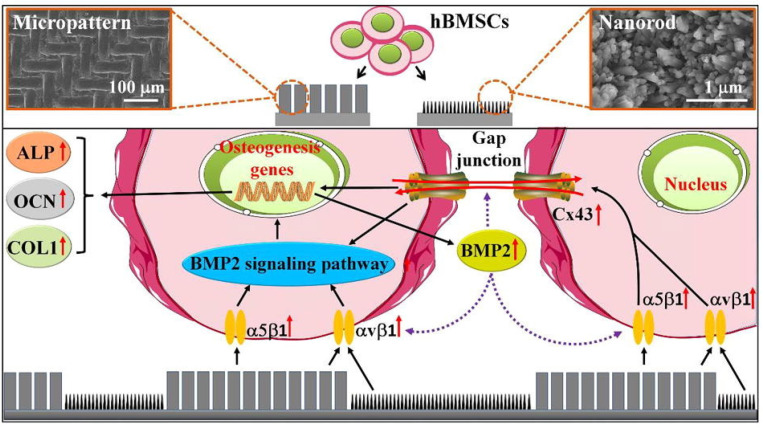
Mechanism of HAP hybrid scaffolds with nano/microstructured surface on the osteogenic differentiation. Reprinted with permission from Ref. [198]. 2018, Zhao, C. C.

**Table 1 jfb-13-00187-t001:** Summary of advantages and disadvantages of CaPs.

	Advantages	Disadvantages	Reference
Species			
HAP	Advanced osteoconductivity and osteoinductivity	Poor mechanical properties, and biodegradability	[29,30,31]
α-TCP	Advanced osteoconductivity and osteoinductivity, easy be resorpoted	Poor stability	[32,33]
β-TCP	Advanced osteoconductivity and osteoinductivity, more stable than α-TCP	Lower biodegradability than α-TCP	[31,34]
ACP	Excellent biodegradability, large specific surface, pH-responsive degradation	Lower surface energy than OCP and HAP, poor stability	[35,36]
OCP	Acts as the initial deposition site for bone, beenter osteoconductivity and osteoinductivity than HAP and ACP	Unstable, poor mechanical properties	[37]
DCPA/DCPD	Good biocompatibility, biodegradability and osteoconductivity	Poor stability, easy to casue inflammatory response by the degradation products	[38,39]
TTCP	Advanced biodegradability, biocompatibility and stability.	Cannot be synthesized in aqueous environment, easy to hydrolysis to HA	[40]
Size			
Microscales	Higher surface charge and excellent molecular adsorption properties	Lower biodegradability	[41]
Nanoscales	Improving the sintering ability of ceramics, and mechanical properties of implants, higher absorbability, easer to penetrate cell membrane.	Difficulty in synthesizing nanomaterials of specific sizes	[41]
Hierarchical nano/micro structures	Similarity of nature bone structure, better cell adhesion ability and bioactivity	Difficulty in controlling	[42]

**Table 2 jfb-13-00187-t002:** The basic information of usual calcium phosphate materials.

Name	Formula	Ca/P	Solubility at 25 °C (g/L)
HAP	Ca_10_(PO_4_)_6_(OH)_2_	1.67	~0.0003
α-TCP	α-Ca_3_(PO_4_)_2_	1.5	~0.0025
β-TCP	β-Ca_3_(PO_4_)_2_	1.5	~0.0005
ACP	Ca*_x_*H*_y_*(PO_4_)*_z_*·*n*H_2_O, *n* = 3–4.5, 15–20% H_2_O	1.2–2.0	/
OCP	Ca_8_(HPO_4_)_2_(PO_4_)_4_·5H_2_O	1.33	~0.0081
DCPA	CaHPO_4_	1.0	~0.048
DCPD	CaHPO_4_·2H_2_O	1.0	~0.088
TTCP	Ca_4_(PO_4_)_2_O	2.0	~0.0007

HAP: hydroxyapatite, α-TCP: α-tricalcium phosphate, β-TCP: β-tricalcium phosphate, ACP: amorphous phosphate calcium. The solubility date at 25 °C of ACP cannot be measured precisely. However, the comparative solubility in acidic buffer is ACP >> α-TCP >> β-TCP >> HAP.

**Table 3 jfb-13-00187-t003:** The commercial products of CaPs.

CaP Materials	Product Name	Producer
HAP	Actifuse	ApaTech, UK
	ApaPore	ApaTech, UK
	Bonetite	Pentax, Japan
	Bone Source	Stryker orthopaedics, NJ, USA
	Bioroc	Depuy-Bioland, France
	Cerapatite	Ceraver, France
	Ostim	Heraeus Kulzer, Germany
	Synatite	SBM, France
β-TCP	adbone^®^ TCP	Medbone, Portugal
	Biosorb	SBM S.A., France
	Cerasorb	Curasan, Germany
	Conduit	DePuy Spine, USA
	Osferion	Olympus Terumo Biomaterials, Japan
	SynthoGraft	Synthograft, MA, USA
	Vitoss	Orthovita, PA, USA
HAP + β-TCP	BCP	Medtronic, MN, USA
	Graftys BCP	Graftys, France
	MBCP	Biomatlante, France
	OsSatura BCP	Integra Orthobiologics, CA, USA
HAP + α-TCP	Skelite	Millennium Biologix, ON, Canada
CDHA	Osteogen	Impladent, NY, USA
ACP + DCPD	Biobon (α-BSM)	Etex, MA, USA
DCPD + β-TCP	ChronOS	DePuy Synthes, PA, USA
TTCP + DCPA + saline	BoneSource HAC	Stryker Instruments, MI, USA
α-TCP + TTCP + CaHPO_4_ + HAP	BIOPEX	Taisho Pharmaceutical, Japan
HAP + collagen	Healos Fx	DePuy Spine, USA
HAP + PLLA	SuperFIXSORB30	Takiron, Japan
HAP + Polyethylene	HAPEX	Gyrus, TN, USA
β-TCP + PMMA	Cal-CEMEX	Tecres Spa, Italy

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
