# Peer review of "Calcium Phosphate-Based Biomaterials for Bone Repair"

_jfb, 2022, doi:10.3390/jfb13040187_

Round 1
Reviewer 1 Report
- English should be improved.
- The report fails to shed light on the future of CaPs; therefore, the authors are suggested to pay more on this critical issue in the text.
- Write something about the computational modeling of CaPs, and its significance in bone tissue engineering.
- Give some examples of CaPs-based products available on the market for bone regeneration
- Some synthesis methods of CaPs including solution combustion synthesis (SCS) are missing.
More descriptions are required about the drug delivery potential of CaPs.
- A table is needed to give an update on in vitro and in vivo experiments of CaPs for bone repair.
- The references should be updated.
Reviewer 2 Report
The manuscript "Calcium phosphate-based biomaterials for bone repair" deals with a very broad subject. In fact the use of calcium phosphates as biomaterials involves many variables to which the present work struggles to give a complete order. Here are some comments:
1- in this paper the most addressed forms of CaPs are crystals and scaffolds, while bone cements and coatings are commented here and there . It would be added value if Authors could focus more just on some forms of CaPs.
2- please, check abbreviations throughout the text: sometimes hydroxyapatite is indicated by HAP, sometimes with HA; DCPA-DCP.
3- English needs careful revision: i.e. line 41 which causing; line 53 physic-chemical and biological properties scaffolds; line 61 provides; line 77 that resulting; line 91 to be investigate; line 110 roles in promote, and so on...
4- line 106- Definition of SPINAL ANIMAL: an animal with its spinal cord cut off from its brain. What do Authors mean?
5- line 143 and 395- DCPD dibasic calcium phosphate dihydrate, not dehydrate
6- line 145- the existence of OCP was definitively ascertained in the 1960s
7- line 168- there are no "molecules" in HA: since it is an ionic compound, 2 "formula units" are contained in the unit cell
8- line 171- it is not correct: structural defects in the synthesized HAP may depend on the synthesis procedure/conditions
9- line 176-177- not all CaP phases are precursors of HA, please add what phases are involved in the phase transition
10- line 179-181- "Equilibrium..." meaning in this sentence is obscure
11- line 186- Sr 2+
12- line 188- ref. 51 this work is not about Mg substituted HA. Literature is rich of important papers dealing with ion substituted HA, that present both physico-chemical and biological properties, please add suitable references and discuss some of them.
13- line 222- what phase of tricalcium phosphate is described here? alpha and beta phases have very different behaviors (also, please check troughout the text in order to distinguish between tricalcium phosphates)
14- line 253-254- Please revise and better clarify: actually pure beta TCP can be obtained from solution, but not alpha TCP. Furthermore beta phase is somehow soluble, indeed undergoes hydrolysis reaction (alpha phase does not).
15- line 266- please distinguish between the phases of TCP: for the most part, beta is used as-is, whereas alpha is used in cements because of its phase conversion into HAP via hydrolysis
16- line 268- what do Authors mean by "rate of absorption"
17- line 280-281- please revise: long range order is a property of crystalline materials
18- line 293- what do Authors mean by "amorphization degree"?
19- line 295- ACP is amorphous and it is not possible to evaluate its degree of crystallinity. Please revise this sentence.
20- line 301- since ACP is amorphous, talking about its structure is misleading: more properly, atom distribution in space
21- line 350- what is "a TTCP: DCPA particle size aspect ratio of approximately 10: 1"?
22- line 369- this conversion time does not occur in body conditions (longer times): please add details about the hydrolisis behaviour of OCP (many interesting papers in literature)
23- line 403- actually it is not so easy to precipitate pure crystalline DCPD: synthesis conditions must be finely tuned. The simplest preparation among CaPs is the one of CDHA.
24- line 428- in this paragraph there is no discussion on "structural characteristics" , but on "particles size". Please change the title accordingly.
25- figure 3 caption- please indicate the crystalline phase in these images, not just CaP
26- lines 622-633- actually the pH influences the equilibria between orthophosphate anions, and related hydrogen-containing anions: please revise this sentence from chemical point of view
27- lines 623-628- this is an extreme simplification: in other cases high temperature produces more controlled dimensions and shapes in comparison to low temperature (see for example the crystallization of HAP from solution). This part should be widened and discussion should enter into details/differences between CaPs.
28- lines 631-633- when using additives as it is described here, the final materials are composite materials that contain also the organic additive: this is out of the scope of this review, the focus is lost and moves to functionalization of CaPs.
29- figure 21 is figure 11; figure 33 is 13
30- line 725- SBF use is very simple and not hazardous in itself, so please revise this sentence that is misleading
31- line 797- if it's a glass, it can not be nanocrystalline by definition
32- line 829- please delete "there are"
33- line 861- "neuralization" is "innervation", I suppose.
34- the entire (short) paragraph 5.3 lacks of focus about CaPs. Looks like it puts into evidence the importance of Mg, which is not the subject of this review. I suggest to cut the paragraph.
35- line 894- "wildly" is "widely"
36- line 904- physiochemical is physico-chemical
37- line 916 and following- point 1 in future perspectives is about composite, hybrid or ion substituted materials, which goes beyond the use of simple calcium phosphates and the scope of this review
38- line 939- please change "remains many challenges" into "remains a challenge"
39- As stated in the conclusion paragraph, lines 900-902, it is correct that "the number of CaP materials with varying species, sizes, and morphologies has mushroomed", but this review fails in putting order and showing the criteria by which "different CaP materials have different properties, which determine the approaches of application"
Reviewer 3 Report
Dear Authors,
Thank you for submitting the manuscript entitled "Calcium phosphate-based biomaterials for bone repair."
This review aims at summarizing the advances in CaP biomaterials with different crystal phase and structure, the strategy for fabricating biomimetic hierarchical nano-/microstructures and highlights their applications in bone regeneration.
The paper is well-written and has several strengths, but there are some critical points that need clarification and improvement.
Introduction
Lines 35-36. Please add “infection” to the list of potential causes of bone defects.
The first part of the Introduction section should be improved, especially regarding the most commonly used methods in treating bone defects. In this respect, authors should mention the use of allografts as an important source of human bone for several orthopaedic applications. Although the potential risk of infections, it is worth noting that allografts are procured, processed, and distributed only by Tissue Banks, which operate under strict guidelines and sterile conditions in Class A environments, i.e., minimizing the abovementioned issues.
Therefore, this reviewer suggests adding this comment in the text.
To this aim, the following reference may help the authors to better focus the field:
Vangsness CT Jr, Wagner PP, Moore TM, Roberts MR. Overview of safety issues concerning the preparation and processing of soft-tissue allografts. Arthroscopy. 2006 Dec;22(12):1351-8.
doi: 10.1016/j.arthro.2006.10.009.
In addition to that, the use of xenografts should be discussed as an alternative treatment option. In this regard, please note that several Companies have in their portfolio commercial products containing bone tissue from different animal sources. Please read the following recent paper to have an overview about commercial allografts and xenografts commonly used by clinicians as current alternatives to autografts:
Govoni M, Vivarelli L, Mazzotta A, Stagni C, Maso A, Dallari D. Commercial Bone Grafts Claimed as an Alternative to Autografts: Current Trends for Clinical Applications in Orthopaedics. Materials (Basel). 2021 Jun 14;14(12):3290. doi: 10.3390/ma14123290.
Lines 60-61. Authors only mention the advantages of synthetic bone scaffolds. However, the well-established poor osteinductive activity and the lack of osteogenic properties of synthetic biomaterials have to be reported.
Methods
Although this work is a narrative review, a brief description of items used to perform the search such as databases, keywords or terms, the years considered, language of papers included in the review, should be provided.
The PRISMA extension for scoping reviews may be helpful to describe the searching items (http://www.prisma-statement.org/Extensions/ScopingReviews).
Therefore, also the abstract should be modified following the new parts added in the methods section.
Figure 1. Please improve the quality of the diagram: since it is a circle, the characters must follow the course of curved lines. Generally, several software (also free) have this function.
Moreover, to make this review more readable, authors should summarize the characteristics, advantages/disadvantages, and relative citing papers in an appropriate table. The latter may better support the information reported in Figure 1.
Finally, I just have a couple of suggestions:
• Harmonize the concepts expressed in the Introduction and Conclusion and Outlook sections, especially considering the comments of this reviewer, and focusing on the biomimetic principle, i.e., a material as similar as possible to the host bone is recommended to allow for the best biological behavior as reported by the following reference:
Boanini E, Gazzano M, Bigi A. Ionic substitutions in calcium phosphates synthesized at low temperature. Acta Biomater. 2010 Jun;6(6):1882-94. doi: 10.1016/j.actbio.2009.12.041.
• Please check the references (preferably with a bibliography software package) and standardised their format in the manuscript.
Round 2
Reviewer 1 Report
The auhtors have well addressed the comments and suggestions. However, Table 4 must be improved by adding a column of remarkable outcomes.
Reviewer 2 Report
The Authors have addressed most of the points raised in the previous review round and the manuscript has been significantly improved.
English language still needs some revision.
Reviewer 3 Report
The manuscript has been sufficiently improved to warrant publication in JFB.
